# Sex differences in intra-set kinematics and electromyography during different maximum repetition sets in the barbell back squat?

**Roland van den Tillaar** *, **Andrea Bao Fredriksen**, **Andreas Hegdahl Gundersen**, **Hallvard Nygaard Falch**

Department of Sports Science and Physical Education, Nord University, Levanger, Norway

* roland.v.tillaar@nord.no.

**Data Availability Statement:** Raw data is included as Supporting information file.

## Abstract

Barbell squats are commonly utilized in resistance training for rehabilitation, daily living enhancement, and improving sports performance. The current study investigated the kinematic and electromyographic (EMG) parameters in the squat between sexes across different repetition ranges (1-, 3-, 6-, and 10-RM) among recreationally strength-trained subjects. A total of 26 subjects (13 men: age 25 ± 3.5 years, height 178.2 ± 5.8 cm, weight 82.3 ± 9.1 kg; 13 women: age 24 ± 4.1 years, height 165.4 ± 6.3 cm, weight 68.2 ± 8.7 kg) participated in the study. The level of significance was set at $p < 0.05$. The findings revealed no sex-specific differences in average barbell velocity across repetition ranges. However, the 1-RM showed a significantly lower average velocity compared to the final repetition of other repetition ranges ($p < 0.001$), with the last repetition at 10-RM revealing a significantly higher velocity ($p < 0.001$). Women had greater maximal angular hip extension velocity in the final repetitions of the 6- and 10-RM ($p \leq 0.035$, $\eta_p^2 \leq 0.20$), while both sexes displayed lower maximal angular knee extension velocity in the final repetition of the 10-RM ($p = 0.028$, $\eta p^2 = 0.15$). Moreover, men had lower EMG amplitude in the rectus femoris (3- and 10-RM), soleus, and lateral vastus (10-RM) compared to women ($p \geq 0.011$, $\eta p^2 \geq 0.26$). It was concluded that 10-RM differed greatly in kinematics and EMG, suggesting different fatigue mechanisms compared to other repetition ranges with heavier loads. Furthermore, sex differences in EMG and angular hip extension velocity might imply sex-specific fatiguing mechanisms during high-repetition squats. These considerations could be important when prescribing training programs.

## Introduction

The squat is a lower-body exercise incorporated into resistance training programs by the general public with the aim of providing health benefits, rehabilitation, and improving daily living, and it is applied by athletes seeking to improve their physical capacity in sports [1]. Many different loading and repetition ranges of the squat have been utilized with the objective of improving physical capacity [2]. For instance, training with high loads and lower repetitions is

**Funding:** The author(s) received no specific funding for this work.

**Competing interests:** The authors have declared that no competing interests exist.

well known to be more specific in improving maximal strength [2], commonly tested in the squat by maximal external load lifted for one repetition maximum (1-RM), due to external load lifted being a result of the maximal force the athlete is able to exert [3]. However, adaptive responses to muscle strength and hypertrophy after a training period are shown to occur at a spectrum of per-set number of repetitions, whereby momentary muscular failure might not be necessitated for improving muscle and strength outcomes [2, 4, 5].

In fact, training multi-joint exercises such as the squat to failure might limit the amount of high-load training volume an athlete is able to complete during a given training period, thus limiting practice specific to 1-RM, as volume and intensity are inversely related [6]. Although maximizing external load is undoubtably specific for improving 1-RM strength in the squat, too frequent training at high loads could result in reduced training volume, physiological stress, and potentially increased risk of injury [7]. As a result of managing training variables such as training volume and intensity of effort, it has become common to utilize self-regulatory strategies such as the RPE scale when training to increase 1-RM strength [8] or more objective methods such as velocity-based training [9]. Independent of training approach, a certain threshold of external load must be acquired for improving 1-RM squat strength, since strength in the squat is load dependent, consisting of skill in the movement, which is dependent on neurological factors [2], constrained by the athlete's anthropometry and capacity (e.g., muscle mass and relative lengths of the limbs) [10].

Technical execution of the squat has furthermore been observed to differ from the start of the set until the end of the set when performing squats with a high repetition range, as human movement is complex and dynamic [11], in which athletes will self-organize joint torques to overcome the load in the squat as momentary failure approaches [12]. Self-organization within a given repetition range might lead to deviations from the specific inter- and intramuscular coordination in 1-RM, as fatigue accumulates throughout a set. So did van den Tillaar and Saeterbakken [13] show that during 6-RM in bench press, the time of the sticking and post-sticking region together with pectoralis and the deltoid muscles increased from the first to the sixth repetition. During 6-RM in squats, muscle activity increased in almost all muscles together with increased lifting time from the first to sixth repetition [14]. Technical breakdown may also vary depending on the repetition range trained, as failure might occur through different neurological, biomechanical, and biochemical mechanisms across different repetition ranges by challenging different physiological (anaerobic and aerobic) systems [15]. For a positive training transfer to occur between a given exercise and performance outcomes, several criteria have been postulated to be of importance [16] (e.g., intra- and intermuscular coordination), whereby deviations in technical execution and limiting factors could reduce the transfer from a given repetition range to 1-RM strength. Inter- and intramuscular coordination involves local muscle specificity. Local muscular specificity is dependent upon muscle action, muscle length during various phases of a lift, similarity in joint kinematics and velocities, all while maintaining an external movement resemblance [16].

Technical variations might possibly be further complicated by differences between the sexes. Men are commonly able to produce more force per unit of mass, which is a result of a populational average of more muscle mass accompanied with less fat mass [17–19], thus expressing greater absolute and relative strength. Women are indicated to possess greater strength endurance abilities at a given relative load [19–22], possibly due to a greater proportion of type I fibers [19, 23], reliance on fat oxidation [20], less oxygen requirements due to less muscle mass [20], and shorter limb lengths reducing work performed per repetition [17]. The mechanism for the differences in strength endurance lacks certainty and has been speculated to vary by strength task being performed [22, 24], as relative strength endurance is suggested to favor women more in the upper limbs and/or tasks at a lower intensity of load [20,

21]. However, these differences are suggested to diminish at higher intensities of load, while electromyography (EMG) has been observed to be similar when normalizing for strength [25].

As such, there exists sex specific repetition ranges that require different demands. Due to the importance of specificity when training for improving 1-RM strength, the objective of the current study was to compare kinematics and EMG between different repetition ranges (1-, 3-, 6-, and 10-RM) and between men and women when performing squats at different repetition ranges to voluntary failure, which viewed synergistically might provide indications of intra- and intermuscular coordinative requirements from the start of a set until reaching voluntary failure. These insights are valuable when designing training programs for improving 1-RM squat strength, if kinematics and EMG can be matched in a sub-maximal set and 1-RM. For the purpose of the current study, it is reasonable to assume that the most important criteria to focus on in terms of specificity are intra- and intermuscular coordination in conjunction with repetition range (external load), as other factors (e.g., proprioception in relation to the environment, direction of force, intention) are assumed to be similar. An increase in load and/or repetitions was hypothesized to correspond to greater EMG amplitude based upon earlier research on the squat [26] and Henneman´s size principle [27]. No difference in EMG amplitude was expected between men and women, as the squat primarily challenges the lower limbs, and all the repetition ranges were performed at a high percentage of 1-RM and normalized to the individual, whereby the recruitment of high-threshold motor units is necessitated towards maximal effort [28]. As a result of self-organization as failure approaches, the kinematics were hypothesized to differ between the initial and final repetitions but be similar when comparing the final repetition across different repetition ranges as this is expected to be a maximal effort until failure which would be similar for all repetition ranges.

## Materials and methods

To investigate the effect of different repetition ranges (1-, 3-, 6-, and 10-RM) on kinematics and EMG amplitude in the barbell parallel back squat exercise, a randomized within-subject, repeated measures design was assessed.

### Subjects

The recruitment of study participants occurred from July 15, 2020, to October 15, 2020. Twenty-six recreationally strength trained men (n = 13) and women (n = 13) volunteered to participate in the study. Inclusion criteria were being able to squat 1.2 x body mass for men and 1 x own body mass for women, with a technique fulfilling regulations set by the International Powerlifting Federation. Further criteria were to declare absence of any injury or illness, which could hinder maximum effort. Furthermore, subjects were instructed not to perform any exercise on the lower limbs and avoid alcohol consumption >48 hours prior to testing. The risks and benefits of participation were explained both orally and in writing, whereby written consent had to be signed prior to participation. The study was approved by the local ethics committee and the Norwegian Center for Research Data (project no. 701688), in conjunction with the latest alteration of the Helsinki Declaration.

### Procedures

To reduce the risk of a learning effect, each subject completed a familiarization session >72 hours prior to the experimental test session. The familiarization followed the same test protocol as the experimental test in order to establish the load for each repetition range. Meanwhile, EMG and kinematic data were only collected in the experimental test session. To increase ecological validity and validity of EMG measurements, stance width was based upon the subjects'

own personal preference and standardized over the different repetition ranges for each subject. However, lifting aids (e.g., lifting belt, knee sleeves) were not permitted, with the omission of lifting shoes. Both sessions started with a standardized warm-up protocol squatting at gradually increasing percentages of 1-RM (40, 60, 70, and 80%) [29], before squatting 1-, 3-, 6-, and 10-RM in a randomized sequence decided by an online randomizer (https://www.random.org). During testing, each subject had a minimum of four minutes of rest between sets to reduce the risk of fatigue influencing performance. At the start of the familiarization session, preferred stance width and body height were acquired with a measuring tape, whereas body mass was measured using a standing scale (Soehnle Professional 7830, stand scale). The barbell was placed on top of the trapezius, also called a high bar back squat [30]. The participants bent from a full knee extension in a self-paced, but controlled tempo until the trochanter major was inferior to the patella; the bottom of the descending phase, which is defined as a parallel squat according to the International Powerlifting Federation's technique regulations, (Fig 1). Appropriate squatting depth was controlled by a 3D motion system to ensure that squat death was consistent among all participants. They then received a verbal signal from the test leader and returned to the starting position. An Olympic barbell (2.8 cm diameter, length 1.92 m) was used in all squats with an with one spotter on each side for safety.

## Measurements

For every subject from each attempt of each repetition, the average barbell velocity during the ascending phase [31] was assessed with a linear encoder sampling at 200 Hz (ET-Enc-02, Ergotest Technology AS, Langesund, Norway), which was attached to the barbell. Eight 3D

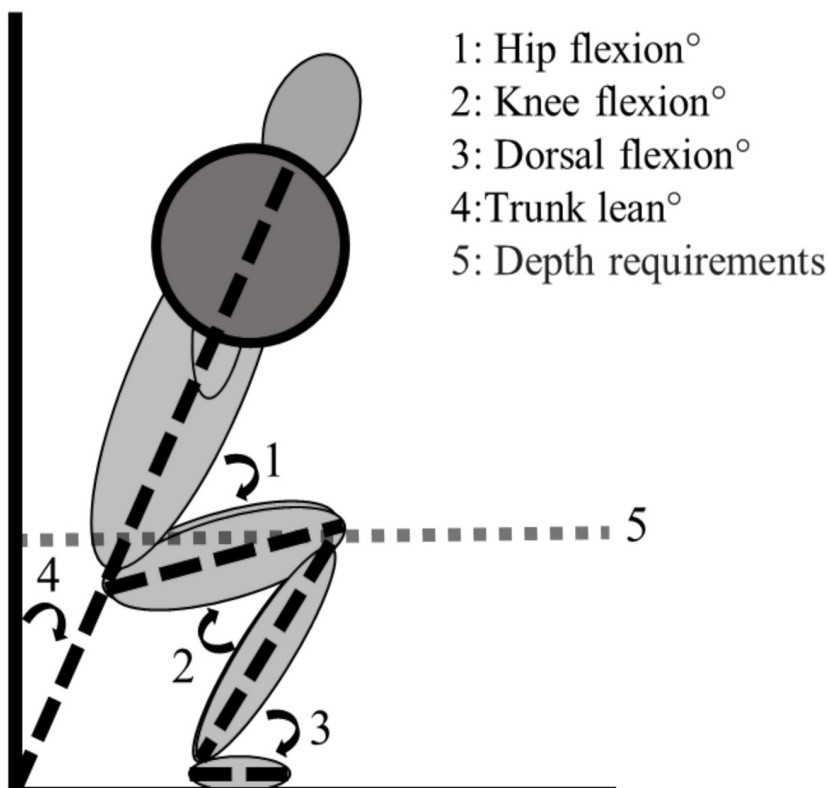

**Fig 1. Depth requirements and definition of joint angles for the parallel squat.**

motion capture cameras (Qualisys, Gothenburg, Sweden) with a frequency of 500 Hz were used to track reflective markers, in order to determine sagittal-plane joint kinematics of the hip, knee, and ankle joints, defined as 0˚ in an erect standing position. The reflective markers were placed at anatomical landmarks on both sides of the body (acromion, pelvis, iliac crest, posterior superior iliac spine, trochanter major, the medial and lateral condyle of the knee, medial and lateral malleolus, sternum, tuber calcanei, 1st and 5th proximal phalanx), creating a 3D measurement of the subjects. In addition, two reflective markers were placed on the lateral tips of the barbell in order to track barbell velocity for synchronizing the linear encoder with the 3D motion capture system. Barbell velocity and trunk lean were defined relative to the laboratory. Maximal angular joint velocities were calculated during the ascending phase. Kinematic data were exported as C3D files to Visual 3D (C-motion, Germantown, USA) for segment building and further analysis.

EMG activity was recorded and analyzed by using Musclelab v.10.200.90.5095 (Ergotest Technology, Langesund, Norway). EMG electrodes (Zynex Neurodiagnostics, Lone Tree, CO, USA), with a sampling rate of 1000 Hz, were placed on the dominant side of 10 different muscles (erector spinae iliocostalis, gluteus maximus, gluteus medius, semitendinosus, bicep femoris, vastus lateralis, vastus medialis, rectus femoris, gastrocnemius, and soleus) Each subject was appropriately shaved, rasped, and cleaned with alcohol prior to electrode attachment in order to reduce noise induced from hair and dead skin. Each electrode (11 mm contact diameter and a 2 cm center-to-center distance) was lubricated and placed lengthways of the presumed direction of the underlying muscle fiber. The location and orientation of the different electrodes were done according to SENIAM recommendations [32, 33]. The root mean square (RMS) of the raw EMG signal was calculated by a hardware circuit network (4th order Butterworth filter: frequency response 20–500 Hz averaging constant 12 ms, total error ± 0.5%). Mean RMS EMG for repetition during the ascending phase was sampled by synchronizing EMG with the linear encoder and normalized for each muscle by the mean RMS activity measured at 1-RM for each muscle.

## Statistical analysis

To assess comparison of barbell velocity development, EMG amplitude, and maximal angular joint velocities over the different RMs and between sexes, a two-way analysis of variance (ANOVA) for sex (independent samples) and repetitions for each RM (3-, 6-, and 10-RM) was performed (repeated measures). In addition, the slope and intercept of the velocity development over the repetitions were calculated for each RM (3-, 6-, and 10-RM) to enable comparison of velocity development across the different RMs and between sexes (2:sexes*3: RMs). Also, the last repetition in each RM was compared between sexes and different RMs (2x4 model). Difference in descriptive and anthropometric measures between the sexes was assessed with the independent samples t-test. Effect size is quantified using partial eta squared ($\eta_p^2$), with values falling within the range of 0.01 to 0.06 considered indicative of a small effect, those between 0.06 and 0.14 representing a medium effect, and $\eta_p^2$ exceeding 0.14 indicating a large effect [34].

## Results

Men were significantly heavier, taller, and of greater absolute strength across all repetition ranges and in terms of relative strength (Table 1). However, no significant differences in percentages of 1-RM at the different RMs was found between sexes (Table 1).

In each RM attempt, a significant decrease in average velocity over the repetitions was found (F≥57.5, p<0.001, $\eta_p^2$≥0.71), while neither significant effects between sexes (F≤1.4,

**Table 1. Mean (±SD) of anthropometrics, absolute and relative strength for men and women.**

| Anthropometrics | | | | |
|---|---|---|---|---|
| | Women (n = 13) | Men (n = 13) | *P* | $\eta_p^2$ |
| Age (years) | 23.9 ± 4.5 | 23.6 ± 1.9 | 0.811 | <0.01 |
| Height (m) * | 1.66 ± 4.5 | 1.81 ± 6.5 | <0.01 | 0.67 |
| Body mass (Kg) * | 63.6 ± 6.6 | 82.2 ± 8.7 | <0.01 | 0.61 |
| Strength | | | | |
| 1-RM (Kg) * | 77.8 ± 12 | 122.8 ± 16.2 | <0.01 | 0.72 |
| 3-RM (Kg) * | 69.1 ± 11.3 | 111.4 ± 14.4 | <0.01 | 0.74 |
| 6-RM (Kg) * | 62.5 ± 9.2 | 92.3 ± 13.8 | <0.01 | 0.71 |
| 10-RM (Kg) * | 56.6 ± 8.2 | 92.3 ± 13.8 | <0.01 | 0.69 |
| Relative strength (1-RM/body mass) * | 1.2 ± 0.2 | 1.5 ± 0.2 | <0.01 | 0.51 |
| % of 1-RM at 3-RM | 88.7 ± 4.4 | 90.7 ± 2.5 | 0.221 | 0.09 |
| % of 1-RM at 6-RM | 80.5 ± 4.5 | 82.4 ± 4.1 | 0.367 | 0.05 |
| % of 1-RM at 10-RM | 72.9 ± 5.5 | 75 ± 3 | 0.301 | 0.06 |

* Indicates significant difference between the sexes on a $p < 0.05$ level.

$p \geq 0.250$, $\eta_p^2 \leq 0.06$) nor interaction effects ($F \leq 1.4$, $p \geq 0.104$, $\eta_p^2 \leq 0.07$) were found (Fig 2). When comparing the last repetition, the average velocity during the 1-RM attempts was significantly lower than during the other RMs, while the velocity of the last repetition was significantly higher at 10-RM than the others ($F = 17.5$, $p < 0.001$, $\eta_p^2 = 0.44$) (Fig 2). No significant differences in average velocity between men and women were found for any of the repetitions at any of the RM attempts ($p > 0.05$). The slope was significantly affected by the RM ($F = 30.5$, $p < 0.001$, $\eta_p^2 = 0.58$), in which the slope became steeper when the RM decreased (10-RM to 3-RM). The intercept increased significantly ($F = 25.1$, $p < 0.001$, $\eta_p^2 = 0.53$) with increase of RM (Fig 2).

When comparing the average velocities between the different RMs (no significant difference between the velocities), it was found that the last repetitions between 3- and 6-RM are similar, and the second repetition of 3-RM was comparable with repetition 5- of 6-RM and repetitions 9 and 10 of 10-RM. The first repetition of 3-RM was comparable with repetitions 3 and 4 of 6-RM for both sexes and repetitions 6–9 of 10-RM for women and 7 and 8 of 10-RM for men. Repetition 2 of 6-RM was again comparable with repetitions 4–6 of 10-RM for both sexes and repetition 1 of 6-RM with 3–5 of 10-RM for women, while for men this repetition was comparable with repetitions 4–6 of 10-RM (Fig 2).

In comparison to the decrease in average barbell velocity over the repetition, the maximal plantar flexion at 3- and 6-RM ($F \geq 2.41$, $p \leq 0.041$, $\eta_p^2 \geq 0.10$) and maximal hip extension velocity at 10-RM ($F = 2.80$, $p = 0.004$, $\eta_p^2 = 0.123$) were significantly affected, while in the other RMs and maximum knee extension, velocity did not significantly change over repetitions ($F \leq 1.89$, $p \geq 0.10$, $\eta_p^2 \leq 0.08$). A significant sex effect was found in maximum hip extension velocity in 6- and 10-RM ($F \geq 5.1$, $p \leq 0.035$, $\eta_p^2 \leq 0.20$), and a significant interaction effect was found in hip extension at 6-RM ($F = 2.4$, $p = 0.04$, $\eta_p^2 = 0.10$). No other significant sex ($F \leq 2.93$, $p \geq 0.10$, $\eta_p^2 \leq 0.12$) and interaction effects ($F \leq 1.56$, $p \geq 0.22$, $\eta_p^2 \leq 0.07$) were found. Post hoc comparison revealed that women had a significantly higher hip extension velocity in 6- and 10-RM than men, and during 6-RM the hip extension velocity decreases, while it stays similar in men over repetitions. The effect of repetitions in plantar flexion occurs mainly in the last repetition, which was higher than the previous one in men (Fig 3), while during 10-RM

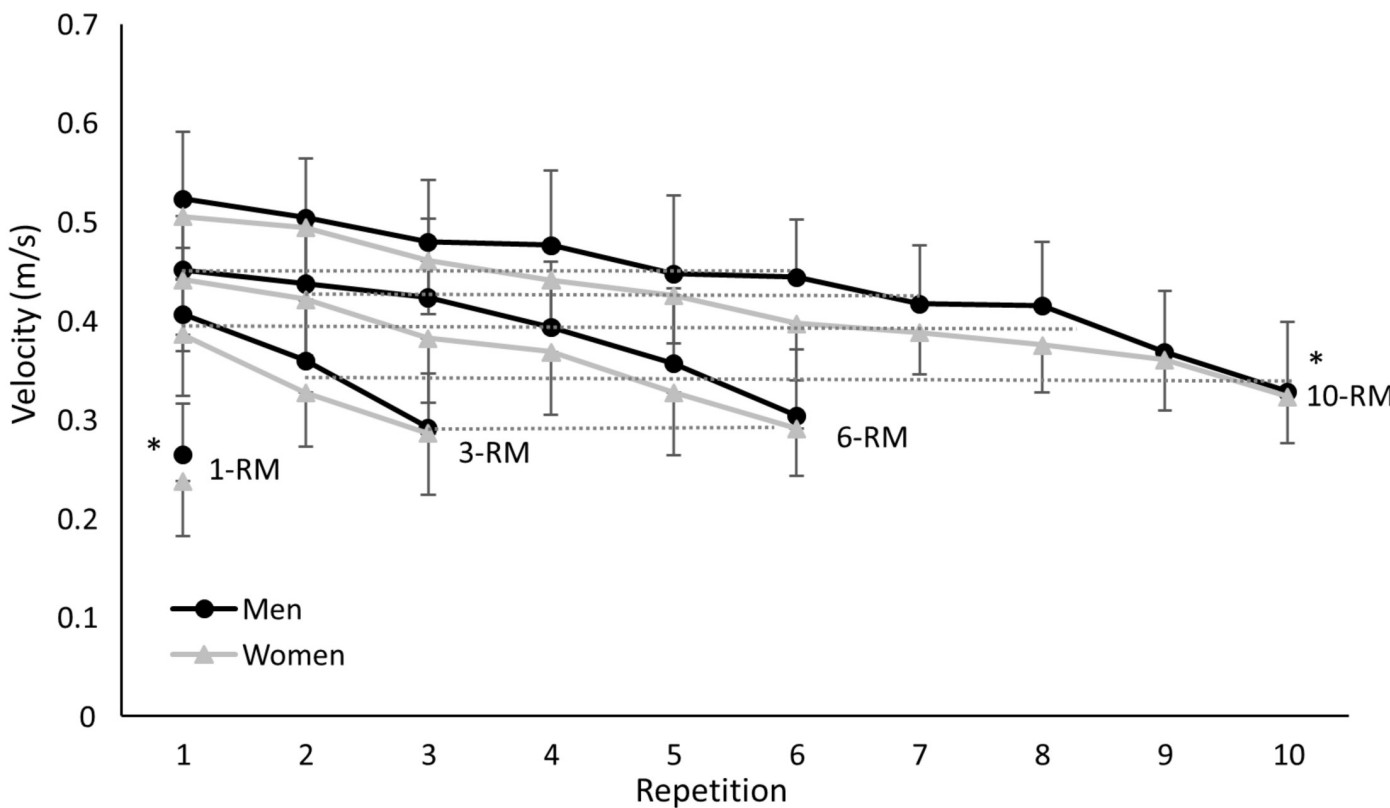

**Fig 2. Average (±SD) velocity averaged over men and women for each repetition during the 1-, 3-, 6-, and 10-RM attempts.** * indicates a significantly different average velocity with all other RMs at a p<0.05 level. Dashed line indicates the corresponding velocities between the different RMs.

the maximal hip extension velocity only in women is significantly lower in the last two repetitions compared to the first two or three repetitions (Fig 3).

When comparing the last repetition, only the maximal angular knee extension velocity was significantly lower at 10-RM compared with 6- and 3-RM in both men and women (F = 3.27, p = 0.028, $\eta_p^2$ = 0.15), while no other significant effects (group, interaction, other joint movements) were found (F≤1.69, p≥0.201, $\eta_p^2$≤0.09) (Fig 3). As minimal differences in maximal angular joint velocities between repetitions were found, also no significant differences in angular velocities were found between the comparable repetitions with the barbell velocities for any of the repetitions and sexes.

In comparison to the peak joint velocities, EMG activity was significantly affected by repetitions for most muscles and RMs (F≥4.465, p≥0.017, $\eta_p^2$≥0.18), except for the soleus and gastrocnemius in any of the RMs (F≤0.83, p≥0.59, $\eta_p^2$≤0.04) and the semitendinosus at 3-RM (F = 0.39, p = 0.68, $\eta_p^2$ = 0.02). A significant sex difference was found for the soleus muscle at 6- and 10-RM (F≥7.8, p≥0.011, $\eta_p^2$≥0.26), rectus femoris at 3- and 10-RM (F≥7.2, p≥0.014, $\eta_p^2$≥0.26), and lateral vastus at 10-RM (F = 11.4, p = 0.003, $\eta_p^2$≥0.34). No significant repetition*sex interactions were found for any of the RMs for any of the muscles (F≤1.72, p≥0.191, $\eta_p^2$≤0.08). Post hoc comparison revealed that the EMG amplitude increased with repetition in most muscles and that men had a lower EMG amplitude than the women in the soleus, rectus femoris, and lateral vastus in these RMs (Fig 4). When comparing the last repetition for the average EMG amplitude, a significantly lower amplitude was measured in the soleus in men for all RMs, and for all quadriceps and calf muscles, the 10-RM compared to the 1-RM. In

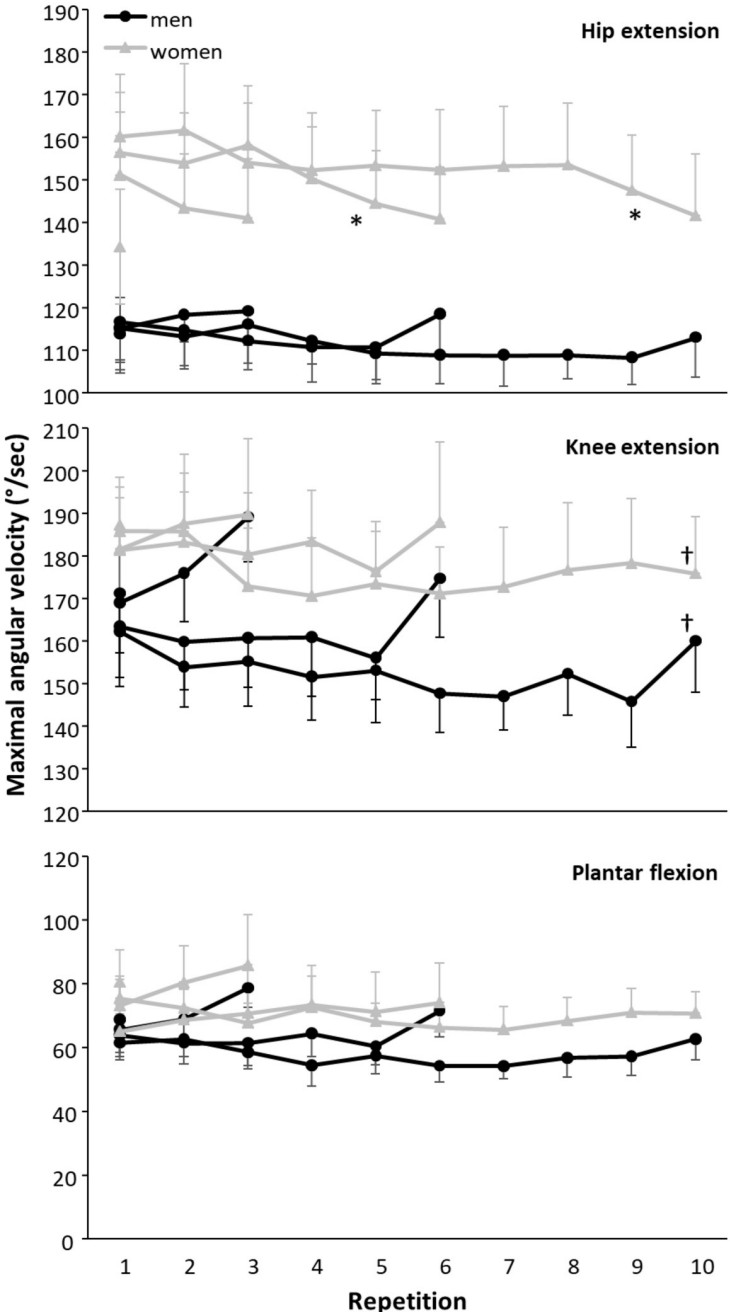

**Fig 3. Average (±SEM) maximal angular hip and knee extension and plantar flexion velocity averaged over men and women for each repetition during the 1-, 3-, 6-, and 10-RM attempts.** * indicates a significantly lower angular velocity for men compared to women for this RM at a p<0.05 level. † indicates a significantly lower knee extension velocity of the last repetition compared with 3- and 6-RM at a p<0.05 level.

women, only the last repetition for the EMG amplitude was lower at 6-RM in the soleus and at 10-RM for the erector spinae (Fig 4). When comparing the repetitions with the same barbell velocity across the different RMs, EMG amplitude is not comparable between these repetitions from the different RMs as the average barbell velocities (Figs 2 and 4).

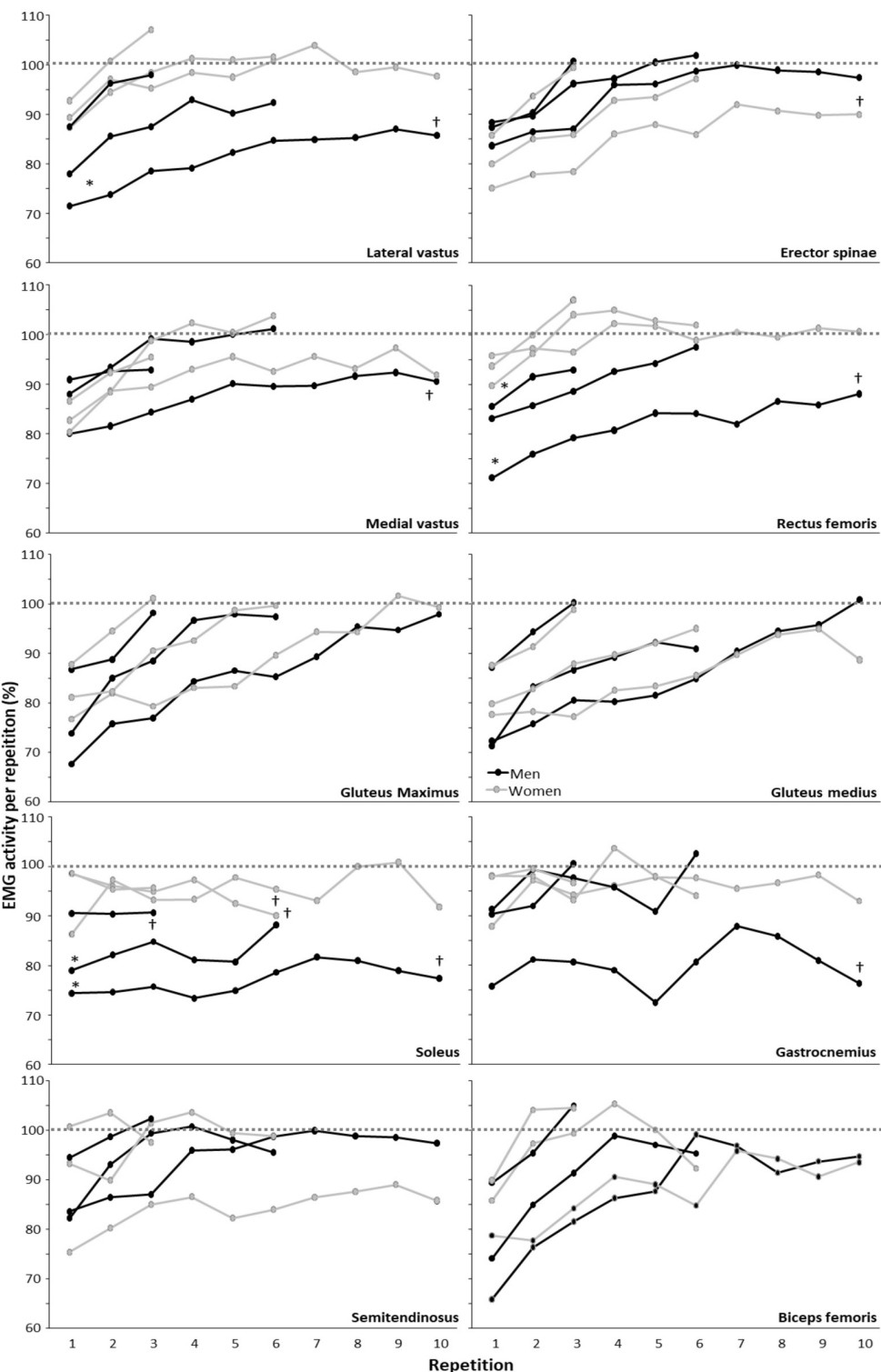

**Fig 4. Average EMG amplitude per repetition normalized to 1-RM for men and women for each muscle during the 3-, 6-, and 10-RM attempts.** * indicates a significantly lower EMG amplitude for men compared to women for this RM at a p<0.05 level. † indicates a significantly lower activity of last repetition compared with 1-RM amplitude at a p<0.05 level.

## Discussion

The objective of the current study was to investigate kinematics and EMG parameters across various repetition ranges (1-, 3-, 6-, and 10-RM) in the squat among recreationally strength-trained men and women. The main findings suggested no sex-specific differences in average barbell velocity over the different repetition ranges. However, average velocity for the 1-RM was significantly lower than the final repetition of other repetition ranges, while velocity of the last repetition at 10-RM was significantly higher. Furthermore, women had significantly greater maximal angular hip extension velocity in the final repetitions of the 6- and 10-RM, while both sexes had lower maximal angular knee extension velocity in the final repetition of the 10-RM compared to other repetition ranges. Men had lower EMG amplitude in the rectus femoris (3- and 10-RM), soleus, and lateral vastus (10-RM) compared to women. Compared to the 1-RM, men had lower EMG amplitude in the final repetition of the 10-RM in the calf and quadriceps, while a similar trend was observed in the erector spinae in women.

Men displayed greater relative and absolute strength, as expected [17], likely due to more absolute muscle mass accompanied by a lower body fat percentage [18, 19]. However, when normalized as a percentage of 1-RM, the 3-, 6-, and 10-RM were not significantly different between the sexes (Table 1). The average barbell velocity development showed a consistent decrease across all repetition ranges (Fig 2), as subjects approached voluntary failure, which was due to accumulated fatigue decreasing their ability to sustain force output, which was expected based upon earlier research [35]. However, this reduction in velocity was not sex-specific, as no significant differences were observed between men and women for any of the repetitions. Nevertheless, the 1-RM revealed a significantly lower velocity, while the final repetition in the 10-RM was significantly higher compared to the last repetition of the 3- and 6-RM ranges (Fig 2). This observation could be due to the effect of the moment of inertia, as previously discussed in a similar study in the bench press [28]. So did Larsen, Haugen and van den Tillaar [28] also observed higher barbell velocity during 10-RM compared with the 1- and 3-RM and since moment arms (distance barbell from shoulder origin) and joint angles were similar between the conditions the moment of inertia would be lower with 10-RM compared to the higher loads. They suggested that a larger moment inertia will increase the time to accelerate the barbell and may have been responsible for the slower barbell velocities observed at 10-RM. and explaining the steepening of the average velocity slope with greater loads.

Furthermore, average velocities were compared between different repetitions across the different repetition ranges. Comparable average velocities during the different repetitions of the different RMs were observed for the following repetitions: rep 2 of 3-RM with 5 and 6 of 6-RM and reps 9 and 10 of 10-RM; rep 1 of 3-RM with reps 3 and 4 of 6-RM and with reps 6–9 (women) and 7 and 8 (men) of 10-RM; rep 2 of 6-RM with reps 4–6 of 10-RM; and rep 1 of 6-RM with reps 3–5 (women) and 4–6 (men) of 10-RM. Thus, based upon average velocity only, several sub-maximal lifts could induce similar requirements despite dissimilar average velocities for the final repetition across the different repetition ranges (Fig 2). This could be important information for training, as a certain average velocity across different repetition ranges could be used as a system to control the training stimulus, similar to the method of repetitions in reserve [8] or percentage of velocity loss [9].

Conversely, the decrease in average velocity over the repetitions seems not to be apparent by a comparable decrease in maximal angular joint velocities of the ankle, knee, and hip individually, as these did not show many changes over the repetitions in the different repetition ranges. It seems that maximal plantar flexion velocity is not influenced much by repetitions, while both maximal knee and hip extension velocity are influenced by the repetitions and RMs. Yet, due to variations between individuals in lifting kinematics (greater hip or knee

extension), no clear strategy over the different repetitions is apparent (Fig 3). However, reduced maximal angular knee extension velocity in the final repetition of the 10-RM (Fig 3) was observed, in comparison to the 3-RM and 6-RM, implying the 10-RM to induce different kinematic requirements, which is in line with the observations in average barbell velocity. Furthermore, women displayed significantly higher maximal angular hip extension velocity compared to men in both the 6-RM and 10-RM squat (Fig 3). This difference might be a result of differences in upper body mass and the length of the trunk. Men had a greater stature and body mass, requiring more force from the hip extensors to accelerate and ascend the barbell and their greater upper body mass [36]. Thereby, the relative contribution of upper body mass to the hip extension velocity will be greater at lighter loads (6-RM and 10-RM), as lighter loads result in lower overall force requirements, making the impact of the upper body mass more pronounced in the context of generating force in the sets with heavier loads (1- and 3-RM) [37]. The difference in maximal hip extension velocity between the sexes might consequently be due to different anthropometrics, to self-organize and reduce the requirements of certain muscles while increasing the contribution of other muscles [12, 38]. Men might perform the squat with less trunk lean/hip flexion, shifting the emphasis more to the knee extensors by reducing the moment arm from the hip to the center of mass [26], which, from a mechanical perspective, reduces work (force x distance) and external torque requirements (load x moment arm) per repetition [17] of the hip extensors.

As average barbell velocity decreases over the repetitions at different RMs, EMG amplitude, as hypothesized, increased with increments in load and/or repetition numbers, with the exception of the calf muscles (Fig 4). That the EMG amplitude did not increase in calf muscles is explainable because the maximal angular plantar flexion did not change much over the repetitions, which indicates that this joint movement does not change much during squats. The increased EMG amplitude with greater loads in squat exercises of the other muscles aligns with prior research [26, 39], as well as when comparing high-load sets to low-load sets as a percentage of 1-RM [40, 41]. However, EMG amplitude did not correspond with the observations of repetitions at similar velocities. Despite similar barbell velocities, dissimilarities in EMG amplitude may imply variations in the muscular environment. When examining sex differences in EMG, men exhibited lower EMG amplitude, particularly in the 10-RM (Fig 4). Further analysis of men revealed a reduction in EMG amplitude in the quadriceps and calf muscles for the 10-RM condition when compared to the 1-RM condition. Reasons for these EMG differences between men and women can be that men tend to accumulate more lactate during high-repetition squats [12]. This possible greater presence of lactate in men is postulated from a higher total proportion of fast-twitch fibers, which might further influence EMG amplitude and the ability to sustain force [42], as earlier research has showed women to possess more fatigue-resistant type I fibers in relative terms [19, 23, 43], making women more reliant on fat oxidation as opposed to men's reliance on glycolytic pathways [20].

Accumulation of lactic acid might also explain reduced knee extension velocity in the last repetition of the 10-RM compared with the last repetition of 3- and 6-RM, as accumulation of lactic acid could particularly affect the knee extensors due to the initial knee extension movement at the first part of the upward phase (pre- and sticking region) in squats [44] after reaching peak hip flexion (lowest barbell height). This was accompanied by differences in time spent under tension (commonly >15 s) and 1-RM (commonly <5 s) [45, 46]. Thus, the set duration in the 10-RM could lead to the failure of higher-threshold motor units, which may be attributed to the accumulation of acid within the muscle cells. This suggestion is supported by Hooper, Szivak [12], who indicated that during high-repetition squats, knee extension decreases when failure approaches. As a result of these physiological changes, there is an increased reliance on lower-threshold motor units to maintain the necessary force output for

completing the final repetition of the 10-RM set [42] as opposed to the 1-RM, where high-threshold motor units are necessitated [27, 47]. Thus, local fatigue in the knee extensors may explain reduced maximal angular velocity in the 10-RM [15].

Conversely, among women, a decrease in EMG amplitude was only evident in the soleus during the final repetition of the 6-RM condition and in the erector spinae during the final repetition of the 10-RM condition in comparison to their respective 1-RM conditions. This could be attributed to the challenges caused by high-repetition squats for women, potentially resulting in greater fatigue in the lower back due to a more "hip-dominant" technique [12] and higher maximal hip extension velocity of the different repetitions than men. This, in turn, could lead to fatigue in the spinal erector muscles, which must sustain force to counteract the gravitational force of the barbell imposed on the back to avoid spinal flexion. About the lower EMG amplitude of the soleus during the final repetition of the 6-RM condition compared to 1-RM could be an indication of different neurological adaptations with different repetition maximums and between sexes. As men in general showed lower EMG amplitudes in the soleus for all RMs, and in the 10-RM compared to the 1-RM for all quadriceps and calf muscles, women did not show this in these muscles. Furthermore, had men a lower EMG amplitude than the women in the soleus, rectus femoris, and lateral vastus in these RMs, indicating another neuromuscular strategy and muscle demand when lifting with these repetition maximums (Fig 4).

In light of the disparities observed in EMG and maximal angular knee extension velocity, it can be speculated that different mechanisms contribute to fatigue in 1-RM versus 10-RM conditions. Therefore, for both sexes, most differences were observed in the 10-RM condition, where EMG amplitude decreased relative to the 1-RM condition, accompanied by a significant reduction in maximal angular knee extension velocity for both men and women. Thus, the 10-RM seems to impose different demands for both men and women in comparison to training with heavier loads. While both sexes display similar average velocities in the final repetition of the 10-RM, they could be constrained by different fatiguing factors, such as differences in muscle fiber type, oxidative metabolism, and alterations in technique related to anthropometric differences [10]. So did Keller, Anders [48] find observed that males and females differ in the underlying strategies to mitigate fatigue while performing a bilateral leg task at 25% of maximum voluntary isometric. They found that men exhibited a faster rate of increase in EMG amplitude, while the women exhibited a slower rate of decline in skeletal muscle tissue saturation.

The current study has limitations that must be addressed. Firstly, the ascending phase was investigated as one phase, while during maximal lifts and when fatigue arises, different regions (pre-, sticking, and post-sticking) occur [49]. The occurrence of these regions could be different at different RMs and be different between sexes. However, this was beyond the scope of the present study and is therefore warranted in future research. Secondly, the study only used EMG as an indirect measure of muscular fatigue, whereby measures of other variables such as lactate or force would strengthen the interpretations. Thirdly, although standardizations such as squat depth, foot placement, and within-day measurements were made to reduce the limitations of EMG, caution is encouraged when interpreting EMG results.

## Conclusions

Based upon the findings of the current study, the 10-RM revealed the greatest difference in kinematics and EMG, which could indicate that subjects reached voluntary failure through other mechanisms than the other repetition ranges with heavier loads. Also, sex differences in EMG for the 10-RM, in conjunction with angular hip extension velocity, might imply sex-

specific fatiguing mechanisms when performing sets of barbell squats in a high repetition range (>6 repetitions) to voluntary failure. As such, the findings highlight the importance of employing lower repetition ranges to mirror strength requirements of a 1-RM squat for both males and females, while 10-RM squat training might present sex-specific demands. Thereby, when conducting lifts with 10-RM in recreational strength-trained persons, athletes and coaches have to be aware that there is a sex difference in neurological strategies resulting in different lifting technique, which could influence the effect of this training differently.

## Supporting information

**S1 Data.**
(XLSX)

## Author Contributions

**Conceptualization:** Roland van den Tillaar, Hallvard Nygaard Falch.

**Data curation:** Roland van den Tillaar, Andrea Bao Fredriksen, Andreas Hegdahl Gundersen, Hallvard Nygaard Falch.

**Formal analysis:** Roland van den Tillaar, Hallvard Nygaard Falch.

**Investigation:** Andrea Bao Fredriksen, Andreas Hegdahl Gundersen, Hallvard Nygaard Falch.

**Methodology:** Roland van den Tillaar, Andrea Bao Fredriksen, Hallvard Nygaard Falch.

**Project administration:** Roland van den Tillaar, Andrea Bao Fredriksen, Andreas Hegdahl Gundersen, Hallvard Nygaard Falch.

**Supervision:** Roland van den Tillaar.

**Validation:** Roland van den Tillaar, Andrea Bao Fredriksen, Andreas Hegdahl Gundersen.

**Writing – original draft:** Hallvard Nygaard Falch.

**Writing – review & editing:** Roland van den Tillaar, Andrea Bao Fredriksen, Andreas Hegdahl Gundersen, Hallvard Nygaard Falch.

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
