## [Decision Letter · Decision Letter 0]

28 Feb 2024

PONE-D-23-33481Sex differences in intra-set kinematics and electromyography during different maximum repetition sets in the barbell back squatPLOS ONE

Dear Dr. Falch,

Thank you for submitting your manuscript to PLOS ONE. After careful consideration, we feel that it has merit but does not fully meet PLOS ONE’s publication criteria as it currently stands. Therefore, we invite you to submit a revised version of the manuscript that addresses the points raised during the review process.

We look forward to receiving your revised manuscript.

Kind regards,

Michał Krzysztofik, Ph.D.

Academic Editor

PLOS ONE

Journal Requirements:

Additional Editor Comments:

**Both reviewers indicate the need for significant revisions of your manuscript, especially the methods section, which requires a more detailed description. **

Reviewers' comments:

Reviewer's Responses to Questions

**Comments to the Author**

1. Is the manuscript technically sound, and do the data support the conclusions?

Reviewer #1: Yes

Reviewer #2: No

2. Has the statistical analysis been performed appropriately and rigorously? 

Reviewer #1: Yes

Reviewer #2: No

3. Have the authors made all data underlying the findings in their manuscript fully available?

Reviewer #1: Yes

Reviewer #2: Yes

4. Is the manuscript presented in an intelligible fashion and written in standard English?

Reviewer #1: Yes

Reviewer #2: No

5. Review Comments to the Author

Reviewer #1: General

The manuscript titled Sex Differences in Intra-set Kinematics and Electromyography During Different Maximum Repetition Sets in the Barbell Back Squat was well written, but revisions are required to fully describe the overall purpose and practical applications of the findings. Firstly, the investigators hypothesized that there would not be a difference between the males and females, yet the Introduction provides background concerning the likelihood of sex differences. Additionally, sex differences are a focal point of the article’s title. Another current major concern is that all figures are currently illegible. Perhaps, there was an issue during the upload. In addition, it is unclear of how barbell velocity was examined. Expanding the methods section associated with that approach should be done. For example, the investigators identified two different techniques of colleting barbell velocity; however, it was not specifically mentioned which was used. Further, what was the rationale of using one, and not the other. Lastly, it is great that the study observed sex differences during various repetitions from various RMs. However, how does the study’s findings translate to usefulness by a coach or the general public looking to exercise? That is, highlighting the practical applications of this study would strengthen the submission. Last general comment includes that sex and gender cannot be used interchangeably. Please choose one and defend why that particular term was used.

Specific

Abstract

The basis for sex difference is not known within the first few sentences. Please introduce the importance of studying the sex difference in the context of this study.

Introduction

Please consider being consistent with the population of interest in the current study. For example, the investigators provided background support of using the 1RM squat test for “athletes” and back squat is used by the general public. However, the current study observed recreationally active individuals. A multiple RM test may be a better approach for participants who are not well resistance trained.

Line 57: Remove and re-define the “x” between volume and intensity

Line 74: It appears that support is missing for the studies objective to measures specific RM’s. Is a citation possible that suggests when fatigue accumulates during a set of an exercise. Please include a citation for the point at which fatigue accumulates. This will help gather support for the study’s methods.

Line 76: Please describe the different physiological systems.

Line 98: Reword sentence for clarity, and define “different demands”. Perhaps, “There exists sex specific repetition ranges that require different demands,” would work.

Lines 111-115: Why was it hypothesized that there would be no sex difference? The title and introduction all point to a clear sex difference.

Lines 116-117: It is not clear on why the investigators would suggest that a similar kinematic would be seen across the final repetition of each different repletion range? When in line 75 it was mentioned that technical breakdown varies across certain repetition ranges. Please clarify.

Methods

Lines 143-144: Please rewrite. It is unclear how squat stance width can be standardized as well as based on the subjects own personal preference.

Line 145: How many of the subjects, and which subjects wore lifting shoes. This may impact the angles seen at the ankle, knee and hip.

Line 155: The use of a 3D motion capture system is great, but was it for reliability? Would it be correct to say to ensure squat death was consistent among all participants.

Lines 160 and 169-170: It is mentioned that barbell velocity was tracked with two different methods. How was one chosen over the other for analysis? Did both methods provide similar outcomes?

Lines 204-206: Re-write for clarity. Perhaps, “Participant characteristics revealed that men were significantly taller, heavier, and displayed greater absolute and relative strength than women. However, men and women did not differ between the parentage of 1RM at different RMs.”

Results

The results are written well, however, in their current state, the uploaded graphs are not readable. The black background hides potentially meaningful information.

Discussion

Lines 310-312: Please further discuss the concept of moment of inertia from a past bench press study, as it relates to your current analysis. That is, please describe citation 27 in the context of this interpretation/finding.

Line 324: I am uncertain of the connection of average velocity to repetitions in reserve. Please further describe the connection between average velocity and RIR. Is this potential relationship well established? Why or why not? Could average velocity be used a validation metric for RIR?

Lines 327-334: Our current understanding is that there were no reported sex differences in barbell velocities across several RMs. Why do the current results differ from all the provided evidence suggesting that there should be a sex difference?

Line 375: The initial knee extension information was found during a 6RM in citation 29 (van den Tillaar et al., 2014) however, your analysis suggests this for the 10RM, but not the 6RM. Why do the current results differ?

Line 376: Was time under tension measured or accounted for?

Lines 388-391: The explanation provided for fatigue associated in the low back is logical. However, further elaboration is needed for the decreased EMG amplitude seen in the soleus muscle.

Lines 400-402: It has been previously shown that males and females differ in the underlying strategies to mitigate fatigue (Keller et al., 2021). Perhaps these previous findings may offer additional interpretations and context.

Keller, J. L., Anders, J. P. V., Neltner, T. J., Housh, T. J., Schmidt, R. J., & Johnson, G. O. (2021). Sex differences in muscle excitation and oxygenation, but not in force fluctuations or active hyperemia resulting from a fatiguing, bilateral isometric task. Physiological Measurement, 42(11), 115004.

Conclusions

Please further explain the practical applications of the sex differences seen during the 10RM.

Figures

All the graphs are illegible. Please uploaded readable figures.

Minor suggestion: Figure 1 - Move the depth requirement line down to identify the greater trochanter being inferior to the patella.

Reviewer #2: Abstract

• The rationale seems inconsistent with the following sections. Please rewrite it.

Introduction

• Overall, the introduction is unnecessarily long and dispersive, and does not provide a solid rationale for the study. I do not understand whether the aim is to examine the sex-differences, as indicated in the title, or different loads, or the influence of fatigue. Additionally, speculation about the possible long-term effects should be avoided. Therefore, I urge the authors reorganizing and simplifying the whole introduction to let me understand why this study is necessary, and what are the possible novelties it may bring.

Methods

• The methods are scarcely described. In first instance, please accurately describe the squatting technique in accordance with a more comprehensive view (DOI: 10.1186/s40798-022-00492-1). Additionally, the EMG description needs to be populated. As examples, please see following references about EMG and squat (DOI: 10.3390/ijerph18020772 and DOI: 10.1080/02701367.2020.1840496). Lastly, while partial eta squared is appropriate for comparing interactions and main factors, pairwise comparisons should be accompanied by another form of effect size, e.g., Cohen’s d.

• Statistical analysis: as the load was assessed separately, I suppose load is not a factor in the design.

There are many points that must be addressed to clarify the manuscript. I will continue the review process once these have been addressed.

Lastly, please be consistent with the taxonomy used and double-check English language. For example, “sex” and “gender” was both used, while it’s clear that “sex” is the only appropriate word.

6. PLOS authors have the option to publish the peer review history of their article (what does this mean?). If published, this will include your full peer review and any attached files.

Reviewer #1: No

Reviewer #2: No

---

## [Author Response · Author response to Decision Letter 0]

21 Apr 2024

Reviewer #1: General

The manuscript titled Sex Differences in Intra-set Kinematics and Electromyography During Different Maximum Repetition Sets in the Barbell Back Squat was well written, but revisions are required to fully describe the overall purpose and practical applications of the findings. Firstly, the investigators hypothesized that there would not be a difference between the males and females, yet the Introduction provides background concerning the likelihood of sex differences. Additionally, sex differences are a focal point of the article’s title. Another current major concern is that all figures are currently illegible. Perhaps, there was an issue during the upload. In addition, it is unclear of how barbell velocity was examined. Expanding the methods section associated with that approach should be done. For example, the investigators identified two different techniques of colleting barbell velocity; however, it was not specifically mentioned which was used. Further, what was the rationale of using one, and not the other. Lastly, it is great that the study observed sex differences during various repetitions from various RMs. However, how does the study’s findings translate to usefulness by a coach or the general public looking to exercise? That is, highlighting the practical applications of this study would strengthen the submission. Last general comment includes that sex and gender cannot be used interchangeably. Please choose one and defend why that particular term was used.

Specific

Abstract

The basis for sex difference is not known within the first few sentences. Please introduce the importance of studying the sex difference in the context of this study.

We have included between sexes to the abstract to introduce what we were investigating.

Introduction

Please consider being consistent with the population of interest in the current study. For example, the investigators provided background support of using the 1RM squat test for “athletes” and back squat is used by the general public. However, the current study observed recreationally active individuals. A multiple RM test may be a better approach for participants who are not well resistance trained.

We have tried to stick to recreational strength-trained subjects, which is also mentioned in the conclusion now.

Line 57: Remove and re-define the “x” between volume and intensity

We have changed it in: volume and intensity are inversely ..

Line 74: It appears that support is missing for the studies objective to measures specific RM’s. Is a citation possible that suggests when fatigue accumulates during a set of an exercise. Please include a citation for the point at which fatigue accumulates. This will help gather support for the study’s methods.

We agree that support is missing as this is one of the main points of the present study since we don’t know if the development is similar over the different repetition ranges. We have included two studies that showed that some changes occur due to fatigue. This is included to the text now: 

So did van den Tillaar and Saeterbakken (13) show that during 6-RM in bench press, the time of the sticking and post-sticking region together with pectoralis and the deltoid muscles increased from the first to the sixth repetition. During 6-RM in squats, muscle activity increased in almost all muscles together with increased lifting time from the first to sixth repetition (14).

Line 76: Please describe the different physiological systems.

We have included anaerobic and aerobic to the text. We have also specified different mechanisms like the biochemical, neurological and biomechanical ones to show that these all can influence fatigue.

Line 98: Reword sentence for clarity, and define “different demands”. Perhaps, “There exists sex specific repetition ranges that require different demands,” would work.

This is changed now in the text.

Lines 111-115: Why was it hypothesized that there would be no sex difference? The title and introduction all point to a clear sex difference.

We have included a ? to the title to question if there are sex differences over the different ranges of repetitions with different loads.

Lines 116-117: It is not clear on why the investigators would suggest that a similar kinematic would be seen across the final repetition of each different repletion range? When in line 75 it was mentioned that technical breakdown varies across certain repetition ranges. Please clarify.

In line 75 we just suggest that it is possible that differences would occur at different repetition maximums. However, this is not investigated before. We think that the final repetition would be the same as this would be a maximal effect just before failure. This is now also mentioned in the text: as this is expected to be a maximal effort until failure which would be similar for all repetition ranges. 

Methods

Lines 143-144: Please rewrite. It is unclear how squat stance width can be standardized as well as based on the subjects own personal preference.

We have rewritten it in: “stance width was based upon the subjects’ own personal preference and standardized over the different repetition ranges for each subject” As it was for each subject individually, but standardized over the different repetition ranges. 

Line 145: How many of the subjects, and which subjects wore lifting shoes. This may impact the angles seen at the ankle, knee and hip.

The main purpose was the effect of different repetition ranges and sex upon kinematics and EMG. We wanted to have the subjects performing in their regular training situation and only a few wore lifting shoes (5 subjects: 2 women 3 men). So we think it does not have a high impact upon the results as it is mainly a within subject design. However, we agree that it could have a small influence upon the joint kinematics.

Line 155: The use of a 3D motion capture system is great, but was it for reliability? Would it be correct to say to ensure squat death was consistent among all participants.

That is correct. We have changed it now in: Appropriate squatting depth was controlled by a 3D motion system to ensure that squat death was consistent among all participants.

Lines 160 and 169-170: It is mentioned that barbell velocity was tracked with two different methods. How was one chosen over the other for analysis? Did both methods provide similar outcomes?

The linear encoder was used for the analysis as that one was automatically synchronized with the EMG system. The markers on the barbel made it possible to synchronize the data with the linear encoder system. This is now also mentioned in the text. “for synchronizing the linear encoder with the 3D motion capture system”

Lines 204-206: Re-write for clarity. Perhaps, “Participant characteristics revealed that men were significantly taller, heavier, and displayed greater absolute and relative strength than women. However, men and women did not differ between the parentage of 1RM at different RMs.”

It is now rewritten in: Men were significantly heavier, taller, and of greater absolute strength across all repetition ranges and in terms of relative strength (Table 1). However, no significant differences in percentages of 1-RM at the different RMs was found between sexes (Table 1).

Results

The results are written well, however, in their current state, the uploaded graphs are not readable. The black background hides potentially meaningful information.

Our excuses for the graphs. We have uploaded the graphs again and now the figures are much better to read, we think.

Discussion

Lines 310-312: Please further discuss the concept of moment of inertia from a past bench press study, as it relates to your current analysis. That is, please describe citation 27 in the context of this interpretation/finding.

We have included some more information about the concept. This is included to the text: So did Larsen, Haugen and van den Tillaar (29) also observed higher barbell velocity during 10-RM compared with the 1- and 3-RM and since moment arms (distance barbell from shoulder origin) and joint angles were similar between the conditions the moment of inertia would be lower with 10-RM compared to the higher loads. They suggested that a larger moment inertia will increase the time to accelerate the barbell and may have been responsible for the slower barbell velocities observed at 10-RM. and explaining the steepening of the average velocity slope with greater loads. 

Line 324: I am uncertain of the connection of average velocity to repetitions in reserve. Please further describe the connection between average velocity and RIR. Is this potential relationship well established? Why or why not? Could average velocity be used a validation metric for RIR?

This is just a suggestion and not studied by us yet. We think it could be a more objective way of controlling training stimulus. However, as said before it is just a suggestion that should be studied more as this is not investigated before and certainly in relation to RIR as that is a more subjective way of training stimulus control.

Lines 327-334: Our current understanding is that there were no reported sex differences in barbell velocities across several RMs. Why do the current results differ from all the provided evidence suggesting that there should be a sex difference?

These sentences show that the change in velocities was not easy visible in the change of angular velocities over the different repetitions and RMs. This was probably due to the individual adaptations. This was not to imply any difference between sexes. We have changed the last sentence in this part to avoid the suggestion. 

Line 375: The initial knee extension information was found during a 6RM in citation 29 (van den Tillaar et al., 2014) however, your analysis suggests this for the 10RM, but not the 6RM. Why do the current results differ?

In the citation only kinematics were analysed around the sticking region of the last repetition of 6-RM and only the timing of the different joint angular velocities were identified. In that study it showed that it was always the knee extension together with the plantar flexion movement that initiated the squat movement and thereby also that this movement could be fatigued earlier as this knee extension every time starts the upwards movement with a large angular velocity, while hip extension is always in the end of the upward movement. As you see in figure 3 it is clear that the Angular velocity decreases during the repetitions, very clearly in men. This also occurs in the 6-RM. However, due to the clear increase in the last repetition and the variability between subjects, this is not statistically possible to show significantly. We only found that the last repetition of 10-RM was lower then the rest for both men and women. However, the last rep of 6-R also had a similar indication compared with 3-RM. This was not significant and thereby difficult to state. We hope we have informed the reviewer more about what we meant with this part of text and we have changed it a little bit to avoid confusion. It is changed in: Accumulation of lactic acid might also explain reduced knee extension velocity in the last repetition of the 10-RM compared with the last repetition of 3- and 6-RM, as accumulation of lactic acid could particularly affect the knee extensors due to the initial knee extension movement at the first part of the upward phase (pre- and sticking region) in squats (43) after reaching peak hip flexion (lowest barbell height). This was …

Line 376: Was time under tension measured or accounted for?

The total time of all lifts could be measured. But we decided not to measure it as we, in general, know that it takes longer time to conduct 10RM compared to 6RM, and 6RM to 3RM. We also did not think that for answering the research question that we should analyze the total time, because we were only interested in the development of the average velocity, which is often used in training, during the different RMs. However, if the reviewer really wants we could calculate that for each attempt for each subject. 

Lines 388-391: The explanation provided for fatigue associated in the low back is logical. However, further elaboration is needed for the decreased EMG amplitude seen in the soleus muscle.

We have elaborated on the soleus in the text now. This is added to the manuscript: About the lower EMG amplitude of the soleus during the final repetition of the 6-RM condition compared to 1-RM could be an indication of different neurological adaptations with different repetition maximums and between sexes. As men in general showed lower EMG amplitudes in the soleus for all RMs, and in the 10-RM compared to the 1-RM for all quadriceps and calf muscles, women did not show this in these muscles. Furthermore, had men a lower EMG amplitude than the women in the soleus, rectus femoris, and lateral vastus in these RMs, indicating another neuromuscular strategy and muscle demand when lifting with repetition maximums (Fig. 4).

Lines 400-402: It has been previously shown that males and females differ in the underlying strategies to mitigate fatigue (Keller et al., 2021). Perhaps these previous findings may offer additional interpretations and context.

Keller, J. L., Anders, J. P. V., Neltner, T. J., Housh, T. J., Schmidt, R. J., & Johnson, G. O. (2021). Sex differences in muscle excitation and oxygenation, but not in force fluctuations or active hyperemia resulting from a fatiguing, bilateral isometric task. Physiological Measurement, 42(11), 115004.

Thank you for this references. We have added some more information to the manuscript about these differences: So did Keller, Anders (48) find observed that males and females differ in the underlying strategies to mitigate fatigue while performing a bilateral leg task at 25% of maximum voluntary isometric. They found that men exhibited a faster rate of increase in EMG amplitude, while the women exhibited a slower rate of decline in skeletal muscle tissue saturation.

Conclusions

Please further explain the practical applications of the sex differences seen during the 10RM.

We have added one sentence more about practical application. However, as we have not conducted a training study we don’t know the longitudinal effect of training with 10-RM between sexes. We have added this to the manuscript: thereby, when conducting lifts with 10-RM athletes and coaches have to be aware that there is a sex difference in neurological strategies resulting in different lifting technique, which could influence the effect of this training differently.

Figures

All the graphs are illegible. Please uploaded readable figures.

We have changed the figures now and uploaded readable figure.

Minor suggestion: Figure 1 - Move the depth requirement line down to identify the greater trochanter being inferior to the patella.

We have the line going through the patella and thereby indicating that the trochanter is inferior of the trochanter. It is exactly the same, but then from the point of the patella. We hope that is ok by the reviewer.

Reviewer #2: Abstract

• The rationale seems inconsistent with the following sections. Please rewrite it.

We have changed some parts of the manuscript and hope that the reviewer is satisfied with the changes.

Introduction

• Overall, the introduction is unnecessarily long and dispersive, and does not provide a solid rationale for the study. I do not understand whether the aim is to examine the sex-differences, as indicated in the title, or different loads, or the influence of fatigue. Additionally, speculation about the possible long-term effects should be avoided. Therefore, I urge the authors reorganizing and simplifying the whole introduction to let me understand why this study is necessary, and what are the possible novelties it may bring.

We have rewritten the introduction aa bit to increase readability. In the present study we wanted to investigate three things: is there a difference lifting technique when using different popular repetition ranges and are these developments comparable over the different repetition ranges. Thirdly, is there a sex difference in these inter- and intramuscular coordination during these different repetition ranges. If there are differences this could have influenc

---

## [Decision Letter · Decision Letter 1]

23 Jul 2024

Sex differences in intra-set kinematics and electromyography during different maximum repetition sets in the barbell back squat?

PONE-D-23-33481R1

Dear Dr. van den Tillaar,

We’re pleased to inform you that your manuscript has been judged scientifically suitable for publication and will be formally accepted for publication once it meets all outstanding technical requirements.

Kind regards,

Emiliano Cè

Academic Editor

PLOS ONE

Additional Editor Comments (optional):

Reviewers' comments:

Reviewer's Responses to Questions

**Comments to the Author**

1. If the authors have adequately addressed your comments raised in a previous round of review and you feel that this manuscript is now acceptable for publication, you may indicate that here to bypass the “Comments to the Author” section, enter your conflict of interest statement in the “Confidential to Editor” section, and submit your "Accept" recommendation.

Reviewer #1: All comments have been addressed

Reviewer #2: All comments have been addressed

2. Is the manuscript technically sound, and do the data support the conclusions?

Reviewer #1: Yes

Reviewer #2: (No Response)

3. Has the statistical analysis been performed appropriately and rigorously? 

Reviewer #1: Yes

Reviewer #2: (No Response)

4. Have the authors made all data underlying the findings in their manuscript fully available?

Reviewer #1: Yes

Reviewer #2: (No Response)

5. Is the manuscript presented in an intelligible fashion and written in standard English?

Reviewer #1: Yes

Reviewer #2: (No Response)

6. Review Comments to the Author

Reviewer #1: Thanks for taking the time to consider my comments/critiques. This was an interesting article to review, read, and consider.

Reviewer #2: (No Response)

7. PLOS authors have the option to publish the peer review history of their article (what does this mean?). If published, this will include your full peer review and any attached files.

Reviewer #1: No

Reviewer #2: No

---

## [Editor Report · Acceptance letter]

26 Jul 2024

PONE-D-23-33481R1 

PLOS ONE

Dear Dr. van den Tillaar, 

I'm pleased to inform you that your manuscript has been deemed suitable for publication in PLOS ONE. Congratulations! Your manuscript is now being handed over to our production team.

Kind regards, 

on behalf of

Prof. Emiliano Cè 

Academic Editor

PLOS ONE